# Establishing an In Vitro System to Assess How Specific Antibodies Drive the Evolution of Foot-and-Mouth Disease Virus

**DOI:** 10.3390/v14081820

**Published:** 2022-08-19

**Authors:** David J. King, Graham Freimanis, Chris Neil, Andrew Shaw, Tobias J. Tuthill, Emma Laing, Donald P. King, Lidia Lasecka-Dykes

**Affiliations:** 1The Pirbright Institute, Woking GU24 0NF, UK; 2Department of Microbial and Cellular Sciences, Faculty of Health and Medical Sciences, School of Biosciences and Medicine, University of Surrey, Guildford GU2 7XH, UK; 3Defence Science and Technology Laboratory (DSTL), Chemical, Biological and Radiological Division, Porton Down, Salisbury SP4 0JQ, UK

**Keywords:** immune escape, foot-and-mouth disease virus, in vitro, virus evolution

## Abstract

Viruses can evolve to respond to immune pressures conferred by specific antibodies generated after vaccination and/or infection. In this study, an in vitro system was developed to investigate the impact of serum-neutralising antibodies upon the evolution of a foot-and-mouth disease virus (FMDV) isolate. The presence of sub-neutralising dilutions of specific antisera delayed the onset of virus-induced cytopathic effect (CPE) by up to 44 h compared to the untreated control cultures. Continued virus passage with sub-neutralising dilutions of these sera resulted in a decrease in time to complete CPE, suggesting that FMDV in these cultures adapted to escape immune pressure. These phenotypic changes were associated with three separate consensus-level non-synonymous mutations that accrued in the viral RNA-encoding amino acids at positions VP2^66^, VP2^80^ and VP1^155^, corresponding to known epitope sites. High-throughput sequencing also identified further nucleotide substitutions within the regions encoding the leader (L^pro^), VP4, VP2 and VP3 proteins. While association of the later mutations with the adaptation to immune pressure must be further verified, these results highlight the multiple routes by which FMDV populations can escape neutralising antibodies and support the application of a simple in vitro approach to assess the impact of the humoral immune system on the evolution of FMDV and potentially other viruses.

## 1. Introduction

RNA viruses such as foot-and-mouth disease virus (FMDV) exist as a genetically heterogeneous swarm within a replication site, where this diversity occurs due to the high replication rate and the poor proofreading activity of the viral RNA-dependent RNA polymerase [1]. The reported error rate is estimated to be between 10^−3^ and 10^−5^ mutations per nucleotide copied [2,3,4], which is hypothesised to result in at least one nucleotide change in every FMDV genome transcribed [5]. While most of this diversity is neutral or negatively affects viral fitness, some mutations can be advantageous, allowing the viral swarm to rapidly adapt to new host environmental pressures and respond to narrow evolutionary bottlenecks [6,7,8].

Specific polyclonal antibodies arising from vaccination or previous infection can prevent virus entry into cells. Viruses often adapt to these immune pressures by generating new variants that can escape these specific responses [9,10]. This mechanism has been shown to drive the antigenic evolution of viruses [11,12], including SARS-CoV-2 [13,14], hepatitis B virus [15] and highly pathogenic avian influenza virus [10]. In the latter example, an in vitro exposure of the virus to a sub-neutralising level of antibodies from a vaccinated chicken led to a total of five consensus-level amino acid substitutions within the hemagglutinin (HA) protein associated with an immune escape phenotype [10].

Foot-and-mouth disease virus (FMDV) serotype A is considered to be an antigenically diverse serotype [16]. In 2015, an FMDV lineage called A/ASIA/G-VII emerged from South Asian countries (such as India, Bangladesh and Nepal) and spread rapidly through countries of the Middle East [17]. The sequencing of field FMDV isolates collected in Iran and Saudi Arabia revealed amino acid differences located within the G-H loop antigenic site of the VP1 protein [17]. These changes were indicative of positive selection, which is hypothesised to have arisen due to incomplete immunity in ruminant populations within the affected regions [18]. While multiple nucleotide substitutions have been identified at antigenic sites, these have been characterised through consensus-level sequencing from a viral population which had undergone multiple rounds of replication through multiple hosts. Previous studies have been undertaken to ascertain the evolutionary mechanisms that occur when a FMDV population is exposed to an immune pressure in vitro. However, these studies did not investigate the evolutionary trajectories of viral populations [19,20]. Therefore, little is known about the processes that drive the generation of immune escape variants at the sub-consensus level in FMDV populations.

To develop an improved understanding of the evolutionary effects of humoral immunity on an FMDV swarm population, an in vitro passage study was established. Here, a field FMDV isolate from the A/ASIA/G-VII lineage was passaged in foetal porcine kidney (LFBK, expressing αVβ6 integrin, a natural receptor for FMDV) cells in the presence of sub-neutralising levels of sera derived from three different settings: (i) cattle which had neither received a vaccine nor been exposed to FMDV (i.e., Control group), (ii) cattle which had been vaccinated with a commercial multivalent FMDV vaccine known to incompletely protect against FMDV A/ASIA/G-VII (i.e., Field group) and (iii) cattle which had received both a vaccine (as above) and subsequent challenge with an FMDV isolate (A/IRN/22/2015) belonging to the A/ASIA/G-VII lineage (i.e., Challenge group). Passaged viruses were isolated and sequenced using a previously optimised high-throughput sequencing (HTS) method [21] to identify non-synonymous substitutions on the surface of the capsid.

## 2. Materials and Methods

### 2.1. Cells

An immortalised line of foetal porcine kidney (LFBK) cells expressing bovine αV and β6 integrin (LFBK-αVβ6 [22]), previously identified as the cellular receptor for FMDV [23], were used in this study. For each viral passage, 10^6^ cells were seeded onto Falcon 6-well clear flat bottom TC-treated multiwell cell culture plates (Scientific Laboratory Supplies, Nottingham, UK) in 2 mL of high glucose Dulbecco’s Modified Eagle Medium (DMEM).

### 2.2. Virus Isolate

The starting material for this study was an FMDV isolate from the A/ASIA/G-VII lineage. This isolate (IRN/22/2015) originated from a vesicular bovine tongue lesion from a field case of foot-and-mouth disease (FMD) collected in Iran and was submitted to the FAO World Reference Laboratory for Foot-and-Mouth Disease (WRLFMD, The Pirbright Institute, UK). The isolate was titrated in bovine thyroid (BTY) cells at a final titre of 10^5^ TCID_50_/_mL_.

In order to establish a stable virus population adapted to the cell, the BTY titrated FMDV isolate was first bulk grown in LFBK-αVβ6 cells and then subsequently passaged five times in LFBK-αVβ6 cells, each time using a MOI of 0.01 plaque forming units (PFU). A MOI of 0.01 was chosen in order to reduce the chance of the formation of defective RNA particles [24]. Following complete virus-induced cytopathic effect (CPE) at each passage, six-well clear flat-bottom TC-treated multiwell cell culture plates (Falcon, Scientific Laboratory supplies) were frozen at −20 °C and then thawed to lyse any remaining cells. This material was centrifuged at 1000× *g* for 3 min at room temperature to pellet cellular debris. The supernatant containing the virus was used as inoculum for subsequent virus passages. To ensure a consistent MOI of 0.01 PFU between each passage, plaque assays were performed for each viral population as described in Appendix B. Following the five passages, the adapted virus became the starting inoculum to be used for viral culture in the presence of FMDV sub-neutralising antibodies.

### 2.3. Sera

Three different sera groups, each containing three samples derived from Holstein Friesian cattle, were used for this study.

Sera from the “Field” group (samples Field Serum 3159, Field Serum 3158 and Field Serum 3817) were derived from uninfected cattle from a farm (which last reported an FMD outbreak in 2008, before the cattle were born) in Saudi Arabia. These cattle were vaccinated with Aftovaxpur^®^ (Boehringer Ingelheim, Pirbright, UK) (four doses, with each at four, five, six and seven months of age [18]. This group was used to represent a vaccine-induced immune response.

The “Challenge” group (samples Challenge Serum 4942, Challenge Serum 4936 & Challenge Serum 4914) consisted of sera obtained from a previously described in vivo vaccine potency study in which cattle (between 6 and 7 months of age) were vaccinated with Aftovaxpur^®^ and then challenged with the FMDV IRN/22/2015 isolate from the A/ASIA/G-VII lineage [25]. These were used to represent both vaccine- and infection-induced immune response. Cattle were challenged 21 days following vaccination. The serum used in this immune escape experiment collected was at the end of the vaccine matching study (i.e., eight days post-challenge [25]). The “Control” group (samples Control Serum 4942, Control Serum 4936 and Control Serum 4914) consisted of sera collected from unvaccinated and non-infected cattle (the same cattle which was used in the Challenge Group, but prior to vaccination and challenge).

The Aftovaxpur^®^ multivalent inactivated vaccine (Boehringer Ingelheim, Pirbright, UK) consisted of FMDV A/SAU/95 and A/IRN/05 antigen components. Since this vaccine provides a partial protection against A/ASIA/G-VII lineage [25], the vaccine was found not to be appropriate for use in an emergency situation (i.e., in previously FMD-free countries) however can be used in endemic countries in which FMDV-susceptible livestock have a degree of protection though practised repeated prophylactic vaccination.

### 2.4. Selection of Sub-Neutralising Levels of Antibodies

In order to allow for viral growth in the presence of immune pressure, the sub-neutralising antibody dilution was determined for each serum from the Field and Challenge groups. To test this, replicates of the starting inoculum were independently exposed to a 10-fold dilution series, ranging from 1/10 to 1/1000 dilutions for the Field group and 1/50 to 1/5000 for the Challenge group. A greater dilution for the Challenge group serum was used, as this group was hypothesised to contain a greater concentration of neutralising antibodies.

Briefly, 100 µL of starting inoculum at a MOI of 0.01 PFU was combined with 1 mL of the required dilution of serum (diluted in DMEM) and incubated at room temperature for 1 h. The virus/serum mix was then added (with the addition of 2.9 mL of DMEM) onto 10^6^ LFBK-αVβ6 cells. Development of CPE was monitored over 48 h using the IncuCyte^®^ Live-cell analysis system (Sartorius, Ann Arbor, MI, USA), with images taken every 30 min using 10× magnification. Usage of the IncuCyte allowed for the replacement of laborious growth curves obtained through plaque assays.

The IncuCyte 2019B Rev2 software was used to analyse the well images to calculate the proportion of cells displaying CPE and determine when complete CPE of all cells had occurred. Due to the poor contrast of healthy LFBK-αVβ6 cells, the analysis module was trained to recognise only unhealthy cells displaying CPE (cells which were rounded and had greater contrast compared to the uninfected control), while healthy cells were ignored. Therefore, the analyser tracked virus infectivity through an increase in CPE over time as opposed to recognising healthy cells and tracking virus infectivity through a decrease in cell confluency over time. Peak CPE was reached when all cells in the well were rounded, as verified by checking the images captured by the Incucyte, with the corresponding cell only control verifying that the CPE was genuine and not an artefact of culture conditions such as overconfluency or exhausted media.

An in-R script (available upon request) was used to process the data, in which the CPE values for each condition were normalised by converting the values to a percentage of the maximum CPE value (where 100% CPE was reached) for each condition. All cell-only controls from the experiments performed here were pooled, averaged and appended with the virus infection data after adjusting the data according to the average normalisation factor of the virus infection data.

### 2.5. Virus Antibody Passage Series

The virus was passaged in LFBK-αVβ6 in the presence of sub-neutralising dilutions of sera as described above. For each serum sample, the virus was continuously passaged four times (P1, P2, P3 & P4), except for Field Serum 3157, where only two passages were performed due to limited volume of serum. After each passage, the virus was titrated, and an MOI of 0.01 PFU was used for inoculation of the subsequent passage. The IncuCyte^®^ Live-cell analysis system was used (as described above) to monitor development of CPE for each passage experiment. The R script described above was then used to normalise the data to a percentage of maximum CPE and to group the data according to sera, with the pooled uninfected cell-only control data appended to each group after multiplying them the average normalisation factor for each group.

### 2.6. Sequencing

Total RNA from viral samples at each passage series was extracted using the TRIzol reagent (Invitrogen) as per the manufacturer’s instructions. FMD viral RNA was then quantified using a specific qRT–PCR assay [26], with a standard RNA generated from the 3D region of FMDV isolate UKG/35/2001. The forward, reverse and probe binding sites targeted by the RT-qPCR assay were confirmed to be identical between IRN/22/2015 and UKG/35/2001 (results not shown). Following this, samples were diluted to 10^6^ FMD viral RNA copies/µL.

Technical duplicates from each RNA sample were reverse transcribed using Transcriptor HiFi polymerase (Roche) following the manufacturer’s instructions with the addition of an oligo dT primer (REV6 [27], GGCGGCCGCTTTTTTTTTTTTTTT) at a final concentration of 0.1 µM, converting FMDV RNA into cDNA. From the cDNA, a 3022 bp product encompassing of the FMDV genomic region, encoding a partial region of the 5′untranslated region (UTR), the leader protein, the capsid proteins (VP4, VP2, VP3, VP1) and the 2A protein was amplified using Platinum SuperFi DNA Polymerase (Thermo Fisher Scientific, Waltham, MA, USA). To reduce the risk of PCR-induced errors, a minimum number of amplification cycles required to produce sufficient amounts of template for sequencing was applied (Appendix A) [21]. The PCR reaction was set up as per the manufacturer’s instructions, using 3 µL of cDNA as the template and each serotype universal FMDV capsid primers (Forward: TGGTGACAGGCTAAGGATG and Reverse: GCCCAGGGTTGGACTC) at a final concentration of 0.5 µM [28], as previously established [21].

PCR amplicons were purified using the Illustra GFX PCR DNA and Gel Band Purification Kit (GE Healthcare, Little Chalfont, UK) as per the manufacturer’s instructions and eluted in 50 µL of nuclease-free water. Quantification was performed using the Qubit^®^ dsDNA High Sensitivity Assay Kit (Thermo Fisher Scientific, Waltham, MA, USA), with samples being diluted to 0.2 ng/µL in nuclease-free water. Library preparation was performed using the Nextera XT DNA sample preparation kit (Illumina, San Diego, CA, USA) and automated on a Hamilton NGS STAR (as per manufacturer’s instructions) in the HTS unit at The Pirbright Institute. Three 2 × 150 sequencing runs were performed on an Illumina MiSeq platform each containing a starting inoculum sample (i.e., the IRN/22/2015 isolated adapted to the LFBK-αVβ6 cells) and a viral passage in the presence of a single serum from the Field, Challenge and Control groups (Appendix A).

Raw reads were deposited into GenBank under the BioProject accession number SUB11548315.

### 2.7. Bioinformatics Analysis

The bioinformatics analyses followed a previously described pipeline [21].

Briefly; the program FastQC (version 0.11.5) [29] was used to assess the quality of raw reads, which resulted in 15 bases and 5 bases from the 5′ and 3′ of each read, respectively, being trimmed using Prinseq-lite (version 0.20.4) [30]. The program Sickle (version 1.33) [31] was then used to remove reads below a qScore of q38 and a length of 70 bp.

To generate a reference sequence, trimmed reads from each of the three sequenced starting inoculum samples were assembled via the de novo assembly program IDBA_UD (version 1.1.1) [32] using default parameters. These output sequences were compared to one another to confirm that they were identical, and were verified using BLASTn [33]. The resulting reference sequence was used to align sequence reads from the passage experiments.

Following this, the alignment program GEM3 (version 1.843) [34] was used to map the trimmed reads of each sample to the reference genome. Output SAM files were converted into BAM files using SAMTools (version 1.2) [35]. BEDTools (version 2.29.0) [36] was used to calculate the coverage depths across the sequenced amplicon. SAMTools mpileup was used to generate a pileup file for each sample, which in turn was passed through an in-house R-script (R version 3.3.1 (available from https://www.r-project.org/, accessed on 18 August 2021)) to determine the frequency of each base at each sequenced position. For sub-consensus single nucleotide polymorphism (SNP) analysis, accepted variant frequency cutoff was based on the number of PCR cycles used during the amplification stage, as previously determined [21]. If a total of 26 amplification cycles was applied to a sample, then a frequency cutoff of 0.2% was used. For samples that required further amplification, a frequency cutoff of 0.5% was applied. Variants occurring in both technical replicates were considered real, and a mean frequency of both was taken. If a technical replicate for a sample failed to produce enough sequence reads, then a higher frequency cutoff of 0.8% was applied to the remaining single replicate. Variants equal and above 50% in frequency were classified as a consensus change.

Primer binding regions used for the PCR amplicon were removed from all analyses.

### 2.8. Mapping Mutations onto A Structure of the FMDV Capsid

In order to further characterise variants found in each sample, non-synonymous substitutions were mapped to the resolved crystal structure of empty capsids of recombinant FMDV A22-H2093F [37] using RasMol (version 2.7.5) [38,39].

## 3. Results

### 3.1. Viral Adaptation to the LFBK-αVβ6 Cell Line

The IRN/22/2015 isolate, which was originally collected from an outbreak in Iran, was submitted to the WRLFMD, where it was initially titrated on BTY cells. In order to focus our study on the effects of the FMDV-specific antibodies on viral populations, the isolate was first passaged in LFBK-αVβ6 cells to adapt the isolate to growth in this cell line.

Firstly, to prepare sufficient viral stock, the IRN/22/2015 isolate was bulk-grown in LFBK-αVβ6 cells (this viral preparation was defined as the “viral stock”), and then subsequently passaged five times in this cell line, each time using a MOI of 0.01 PFU. To assess the adaptation of the IRN/22/2015 isolate to the LFBK-αVβ6 cells, growth of the virus after each passage in LFBK-αVβ6 cells was assessed with the assumption that an adapted virus would show no further growth acceleration on subsequent further passages. In order to monitor viral growth in real time, the IncuCyte^®^ (Sartorius, Ann Arbor, MI, USA) Live-cell microscopy system was used, and the appearance of CPE (manifested as rounding cells) was monitored every hour post-infection (Appendix A, Figure 1).

While the time to complete CPE (i.e., the initial time when all cells are rounded) for the virus stock was 31 h post infection (hpi), this time decreased to 17, 14 and 12 hpi after one, two and three passages in LFBK-αVβ6 cells, respectively. Following the fourth and fifth viral passages, there were no further reductions in the time to complete CPE (13 and 12 hpi, respectively), indicating that the virus was adapted to the cell line (Figure 1).

### 3.2. Viral Adaptation to Sub-Neutralising Levels of FMDV-Specific Antisera

To better understand the evolution of antibody escape mutations within FMDV swarm populations, the IRN/22/2015 isolate was passaged in the presence of sub-neutralising sera obtained from either vaccinated cattle (Field group) or vaccinated and then challenged cattle (Challenge group).

To determine an optimum sub-neutralising level of antibodies, samples from the Field and Challenge groups were subjected to a 10-fold titration experiment, whereby the growth delay of the FMDV isolate was assessed. These preliminary studies indicated that 1/10 and a 1/50 dilutions for the sera from the Field and Challenge groups, respectively, were the optimal dilutions to establish sub-neutralising conditions, while sera at higher concentrations fully inactivated the virus. For the sera from the Control Group, a 1/10 dilution of each serum was used.

During the first passage of the virus with the sub-neutralising sera the development of the virus-induced complete CPE was delayed. The only exception was the virus grown in the presence of Field Serum 3157, for which the time to complete CPE was similar to that observed for the virus grown in the presence of Control sera (i.e., 14 hpi, indicating none or limited impact of the sera on the time taken to generate CPE). Overall, viruses grown in the presence of sera from the Challenge group induced complete CPE; at the initial passage, between 24 and 44 (mean 33.33) hpi, viruses grown in the presence of Field Serum 3159 and Field Serum 3817 showed complete CPE after 23 and 25 hpi, respectively, while viruses grown in the presence of Control sera showed a complete CPE after 10 to 13 (11.33) hpi (Figure 2).

From the second passage onwards, a trend of reduction in the delay of time to complete CPE was observed for viruses grown in the presence of samples Challenge Serum 4942 and Field Serum 3159 (Figure 2). This trend, albeit to a lesser extent, was also observed for the viruses grown in the presence of serum samples Challenge Serum 4926, Challenge Serum 4914 and Field Serum 3817, with a significant reduction in time to complete CPE observed between passage 4 and 1 for viruses passaged in presence of the Challenge sera (Figure 2). While some reduction in time to CPE was also observed for virus grown in the presence of Control Serum 4926, this was to a significantly lower extend that to the one observed for Challenge sera. Yield of the passaged viruses ranged between 2.10 × 10^6^ PFU/mL to 8.00 × 10^7^ PFU/mL (Appendix A).

### 3.3. Identification of Mutations Acquired during Viral Passage in the Presence of FMDV Sub-Neutralising Sera

To investigate mutations which contributed to the adaptation of the viral populations to the antibody pressure, RNA from each viral passage was sequenced using the Illumina MiSeq. It was hypothesised that mutations facilitating immune escape will be localised mainly in the region encoding the capsid proteins, and therefore, a genomic fragment which started at the end of the 5′ UTR and spanned to the region encoding the 2A protein was amplified and sequenced. To control for errors introduced during reverse transcription, PCR and sequencing, each RNA sample was processed as technical duplicates. Apart from one technical duplicate of a virus population from passage 2 grown in the presence of serum sample Field 3817 (which failed to sequence due to an unknown technical reason and therefore was removed from further analysis), all samples yielded sufficient coverage depth, between 1.31 × 10^3^ and 8.88 × 10^4^ (mean of 1.37 × 10^4^) nucleotides per position across the sequenced amplicon (Appendix A). Following sequencing, between 1.13 × 10^5^ and 8.34 × 10^5^ (mean of 3.72 × 10^5^) reads were produced for each sample. Of this, between 75.20% and 99.52% (mean of 97.98%) sequencing reads aligned to the reference genome. Appendix A summarise the RT-qPCR, PCR and sequencing data, respectively, for all samples.

Finally, regions of primer binding were removed from further analyses, and only mutations which were above a previously determined frequency threshold (i.e., 0.2–0.8% depending on the amount of starting material; see Material and Methods for details) were reported in this study. To be able to determine mutations specific to the treatment with FMDV sub-neutralising sera, starting inoculum (i.e., IRN/22/2015 isolate adapted to LFBK-αVβ6 cells) and viral populations passaged in the presence of the Control sera were also sequenced.

Not surprisingly, in all samples, mutations at low frequency were most numerous (Figure 3 and Appendix A). While the starting inoculum had on average 26 mutations, this number was observed at similar levels in other samples (i.e., 20). The average frequency of mutations within the starting inoculum was 1.57% (0.20–13.82%) (Figure 3 and Appendix A). For comparison, mutations found in viral populations passaged in the presence of Control sera had an average frequency of 4.73% (0.20–58.64%), while mutations found in viral populations passaged in the presence Field sera had an average frequency of 7.01% (0.21–93.84%)—in the presence of Challenge sera, 5.84% (0.23–97.07%).

There were three non-synonymous substitutions found in the region encoding the Leader protein (L^pro^) which were either specific to treatment with FMDV sub-neutralising sera or increased upon passage in the presence of sub-neutralising sera and which, once detected, persisted in all subsequent passages (Figure 3 and Table 1). K29R mutation was present at low frequency in the starting inoculum and all passage 1 samples; however, its frequency drastically increased over passages in the Field Serum 3159, reaching consensus at passage 3 and frequency of 74.39% at passage 4 (Figure 3 and Table 1). While the N189D mutation stayed at low frequency at all four passages, it was specific to the viral population passaged in the presence of Challenge Serum 4926, with no mutation detected at this position in the starting inoculum or viruses passaged in the Control sera (Table 1, Appendix A). Finally, although the W192R mutation was present in the starting inoculum at a low frequency (1.96%), the frequency of this mutation drastically increased with each passage in both Control sera and the FMDV sub-neutralising sera. Interestingly, this increase in frequency was more rapid for the FMDV sub-neutralising sera, with the W192R mutation reaching consensus already at passage 2 for some samples and at passage 3 for the other samples, while in the Control sera, the consensus was reached at passage 4 (Table 1).

Six non-synonymous mutations, which were either specific to the treatment with FMDV sub-neutralising sera or drastically increased in frequency during passage in the presence of FMDV sub-neutralising sera and which continued over the subsequent passages, were detected in the region encoding the capsid protein (Table 1). The region encoding the VP4 protein contained a single (L71P) mutation. This was detected at low frequency in viral populations treated with Field Serum 3159 and Challenge Serum 4926; however, it reached a frequency of 40.07% over four passages only in the viral population grown in the presence of Challenge Serum 4942 (Table 1). Two non-synonymous changes specific to FMDV sub-neutralising sera were found in the region encoding the VP2 protein (L66F and K80E). The L66F mutation was detected only in the viral population grown in the presence of Field Serum 3817 and reached consensus-level (60.95%) at passage 4 (Figure 3 and Table 1). Although the K80E mutation was detected as a minor variant (0.44%) in the starting inoculum, it reached consensus-level only in the viral population grown in the presence of Field Serum 3159 (Table 1). A single mutation, which was specific to FMDV sub-neutralising sera, was found in the region encoding the VP3 protein. While this mutation (P4S) was specific to the virus population grown in the presence of Challenge serum 4942, it did not reach consensus, with the highest frequency of 9.89% found in passage 2. Two mutations specific to FMDV sub-neutralising sera were detected in the region encoding the VP1 protein (Q58H and A155T). The Q58H mutation was specific to treatment with Challenge Serum 4926, however it remained as a minority variant throughout all four passages (Table 1). The A155T mutation was identified at consensus-level by passage 3 in the viral population grown in the presence of Challenge Serum 4942. However, the same mutation was also detected in passage 4 of the viral population passaged in the presence of Field Serum 3817 as a low-frequency variant (1.68%) (Table 1).

Only a single synonymous mutation which reached consensus, S37 located in the region encoding the VP2 protein, was found in the entire amplicon. This mutation was found in the starting inoculum at frequency of 13.17%, oscillated from 9.23% to 23.68% in the viral populations grown in the presence of the Control sera and reached consensus level in passage 3 in the viral population grown in the presence of Field Serum 3159. Although the S37 mutation was also found in viral populations grown in other FMDV sub-neutralising sera (i.e., Field Serum 3157, Field Serum 3817, Challenge Serum 4914, Challenge Serum 4926, and Challenge Serum 4942), in these samples, the frequency of the S37 mutation reduced with each passage (Table 1). In addition to the mutations described above, numerous transient nucleotide substitutions were detected within all viral populations. These mutations were present at sub-consensus frequencies (Appendix A).

Overall, five consensus (VP2^37^, VP2^66^, VP2^80^, VP1^155^) or near-consensus (VP4^71^) mutations which were specific for the treatment with the FMDV sub-neutralising sera were found in the viral populations (Table 1), with the highest proportion occurring within the VP2-encoding region.

### 3.4. Non-Synonymous Substitutions Exposed on the FMDV Capsid Surface

To investigate which mutations observed during the viral passage series could contribute to the viral evasion of neutralising antibodies, all non-synonymous mutations which were specific to the FMDV sub-neutralising sera were mapped onto the FMDV capsid using the crystal structure for the capsid of FMDV A22-H2093F. This isolate was the closest FMDV isolate for which a capsid structure prediction was available. The amino acid sequences of A/IRN/22/2015 and A22-H2093F were compared and were found to have 100% identity in the region encoding the VP4 protein, 93% identity in both the regions encoding the VP3 and VP2 proteins and 91% identity in the region encoding the VP1 protein.

From the six non-synonymous mutations detected in the region encoding the capsid proteins and described above, three were exposed on the surface of the viral capsid. These were located at VP2^66^, VP2^80^ and VP1^155^. For the VP2^66^ substitution from leucine (L) to phenylalanine (F) led to an accommodation of a larger aromatic side chain, while keeping site’s hydrophobic properties. Amino acid substitution from lysine (K) to glutamic acid (E) at position VP2^80^ led to a change of the positively charged side chain to a negatively charged side chain. The substitution of alanine (A) to threonine (T) at position VP1^155^ led to a change from a hydrophobic side chain of to a polar, uncharged side chain. In both latter cases, the change in the side chain length was only of a single carbon.

## 4. Discussion

This study was designed to investigate the consensus and sub-consensus nucleotide variation within FMD viral populations exposed to sub-neutralising dilutions of virus-specific antisera. A FMDV A/IRN/22/2015 isolate (of the A/ASIA/G-VII lineage) was passaged in LFBK-αVβ6 cells in the presence of three different serum conditions, representing no immune pressure (Control group), vaccine-only-induced pressure (Field group) and immune pressure following vaccination and challenge (Challenge group). Three sera for each of these conditions were tested independently.

To limit the detection of nucleotide substitutions which were induced by viral adaptation to the LFBK-αVβ6 cells, the A/IRN/22/2015 isolate was initially passaged five times in this cell line. After the third passage of the A/IRN/22/2015 isolate in the LFBK-αVβ6 cells there was no increase in the rate of replication, suggesting that the virus adapted to growth in the LFBK-αVβ6 cell line. While the viral population of the A/IRN/22/2015 isolate was passaged five times in the LFBK-αVβ6 cells, sub-consensus nucleotide substitutions were present within the starting inoculum. A number of low-frequency substitutions were expected in the adapted virus due to the poor proof reading ability of the viral-RNA-dependent RNA polymerase [5]. While a clonal population could have been used for this study, this would not have mimicked a natural infection and would also suffer from mutations which resulted from population expansion. It is possible that during viral adaptation to the LFBK-αVβ6 cell line, mutations at higher frequencies occurred elsewhere in the genome. These would have been missed by this study, which focused on mutations occurring in the capsid-encoding region.

While no measurable delay was observed in the time taken to generate complete CPE for the virus populations grown in the presence of Control sera (i.e., the time taken to generate the complete CPE was similar to the time taken by the starting inoculum), treatment with Field and Challenge sera caused a delay in FMDV-induced CPE. The evolutionary impacts of these immune pressures on swarm diversity were monitored over sequential passages in which a fixed dose of the FMD virus was cultured in the presence of the same sera. Any reversion of the inhibitory effects of the antisera was hypothesised to be associated with the evolution of the starting inoculum to escape these immune pressures. Such reversion in the inhibitory effects of the antisera over passages was the most apparent for the virus grown in the presence of Challenge Serum 4942 and/or Field Serum 3159.

A small reduction in time to CPE was also observed for virus grown in the presence of Control Serum 4926, indicating that the virus adapted to some components of bovine serum, such as cytokines, which could vary in level between animals. The levels of cytokines in the sera were not measured due to a limited volume of sera, which was sufficient only for a single experiment. Serum from the Challenge group collected eight days post-challenge would likely have upregulated anti-viral cytokines [40], however, since FMDV is known to be sensitive to interferons (IFNs) [41,42]; it was assumed that if any significant amount of IFNs was present in any sera, it would abort FMDV replication, especially since a very low MOI (0.01 PFU) was used in this study. Finally, this reduction in time to CPE observed in response to passage in the presence of Control Serum 4926 was less striking that the reduction observed for the virus passaged in the presence of Challenge Serum 4942 or Field Serum 3159.

For three of the virus–serum experimental conditions, unique consensus-level non-synonymous substitutions that were predicted to change amino acid residues expressed on the surface of the FMDV capsid (VP1^155^, VP2^66^ and VP2^80^ within viral populations passaged in the presence of serum samples Challenge 4942, Field 3917 and Field 3159, respectively) were detected. These changes were located close to previously characterised antigenic sites. The VP1^155^ site is adjacent to a substitution within antigenic site one (VP1^138−154^) identified in field isolates from the A/ASIA/G-VII lineage recovered from vaccinated cattle in Saudi Arabia [17], while the change at VP2^66^ is adjacent to an antigenically variable site at VP2^64^ found in serotype A FMDV collected from the Middle East [43]. The VP2^80^ substitution, which was also identified at a low frequency within the starting inoculum and the virus population exposed to the Control Serum 4942 or Field Serum 3157, has been previously characterised as an epitope region within FMDV serotype A10 [44]. Furthermore, nearby antigenic sites have been identified at VP2^88^ in serotype A viruses [45], VP2^79^ in A5 viruses [46] and VP2^82^ in A22 viruses [47] circulating in East Africa.

These three unique consensus-level substitutions observed in three samples could be due to variation in antibody composition of the serum, whereby the respective host developed a subtly different spectrum of immune responses against the different antigenic determinates of the virus, which translated to a different evolutionary response for the FMDV isolate. An alternative theory could be that the evolution of FMDV is non-deterministic in nature, with virus swarms exploring many surface-exposed amino acid substitutions that might provide a potential solution for immune escape. Future repeat studies using biological replicates of the same virus and serum pairs are required to characterise the repeatability of mutations developed by FMDV in response to antisera. In addition, future studies could use the three serum conditions (control, vaccinated and unchallenged and vaccinated and challenged) from the same animals to limit the impact of host immune variability.

Viruses passaged in the presence of Challenge Serum 4914, Challenge Serum 4926 and Field Serum 3157 failed to generate any consensus-level surface-bound non-synonymous mutations. A possible reason for this could be that the levels of neutralising antibodies in antisera from these serum samples were not sufficient to induce further evolution in the viral swarm. Alternatively, the constraints arising from the transmission bottleneck between each passage could negatively affect viral evolution. By using a starting MOI of 0.01 PFU, the genetic diversity in the founder population between passages will be limited [8]. FMDV populations exposed to a lower concentration of sub-neutralising antibodies may generate immune escape variants at lower frequencies, which when diluted to a MOI 0.01, may not be present within the founder population of the subsequent passages, limiting the chance of an immune escape substitution reaching consensus level. A previous study investigating the bottleneck effect in FMDV populations concluded that when a narrow transmission bottleneck is present, the higher frequency substitutions generated in the initial passage have a greater chance of becoming established at consensus level [8]. This is in line with what was found in this study, where high-frequency surface-exposed substitutions identified in the initial passage of virus populations grown with serum samples Field Serum 3817, Field Serum 3159 and Challenge Serum 4942, (Table 1 and Figure 3) were also present (at a higher frequency) in the subsequent passages.

In addition to surface-exposed amino acid substitutions, there were three amino acid mutations (L71P in VP4, P4S in VP3 and Q58H in VP1) that were specific to the treatment with the FMDV sub-neutralising sera but found at locations on the surface of the capsid at which they were not predicted to appear. Two of these (at locations VP4^71^ and VP^34^) were found in the viral population grown in the presence of Challenge Serum 4942, the viral population which showed a reversion in anti-sera-induced delay in the occurrence of CPE through a potential immune escape substitution at VP1^155^. While the N-terminus of the VP4 protein in other picornaviruses is known to be transiently exposed to the outside of the capsid during a process called virus breathing [48,49,50], it is not clear whether the C-terminus of the VP4 protein is exposed to the antibodies. Finally, a single synonymous substitution (S37) was found in the region encoding the VP2 protein. While this substitution was present at a low frequency in the starting inoculum and in samples treated with Control sera, it rose in frequency, reaching the consensus in the viral population passaged in the presence of Field Serum 3159, the viral population which showed a reversion of the delay in time to CPE. Further studies are required to verify the phenotype of these mutations.

Outside of the region encoding the capsid proteins, three non-synonymous substitutions (K29R, N189D and W192R) were found in in the region encoding the L^pro^. Interestingly, the W192R mutation was found in the starting inoculum and increased in frequency over the passages in all viral populations, reaching consensus at a faster rate in the viral populations grown in the presence of the Field and Challenge sera. It is possible that this mutation occurred due to continued viral adaptation to the LFBK-αVβ6 cells, which was not reflected by enhanced CPE. L^pro^ is a protease catalysing, which self-cleaves from the FMDV polyprotein and is involved in the cleavage of host factors halting translation of host proteins and innate antiviral responses [51,52]. When comparing to variants with other amino acids at the L^pro192^ position, tryptophan was related to an enhanced self-processing by L^pro^ [53]. It is possible that the enhanced self-processing by L^pro^ was no longer required in the LFBK-αVβ6 cells, resulting in the A/IRN/22/2015 isolate losing this phenotype. However, without functional studies, it is difficult to interpretate the role of these non-capsid mutations.

This study highlights the potential of using an in vitro system to characterise the evolution of immune escape variants under different serum conditions. Three potential surface-bound immune escape substitutions were characterised within the VP2 and VP1 encoding regions, Overall, this study indicates that FMDV adopts alternative evolutionary routes to escape immune pressure, and this in vitro system has a potential to be tailored to studying the evolution of other viruses.

## Figures and Tables

**Figure 1 viruses-14-01820-f001:**
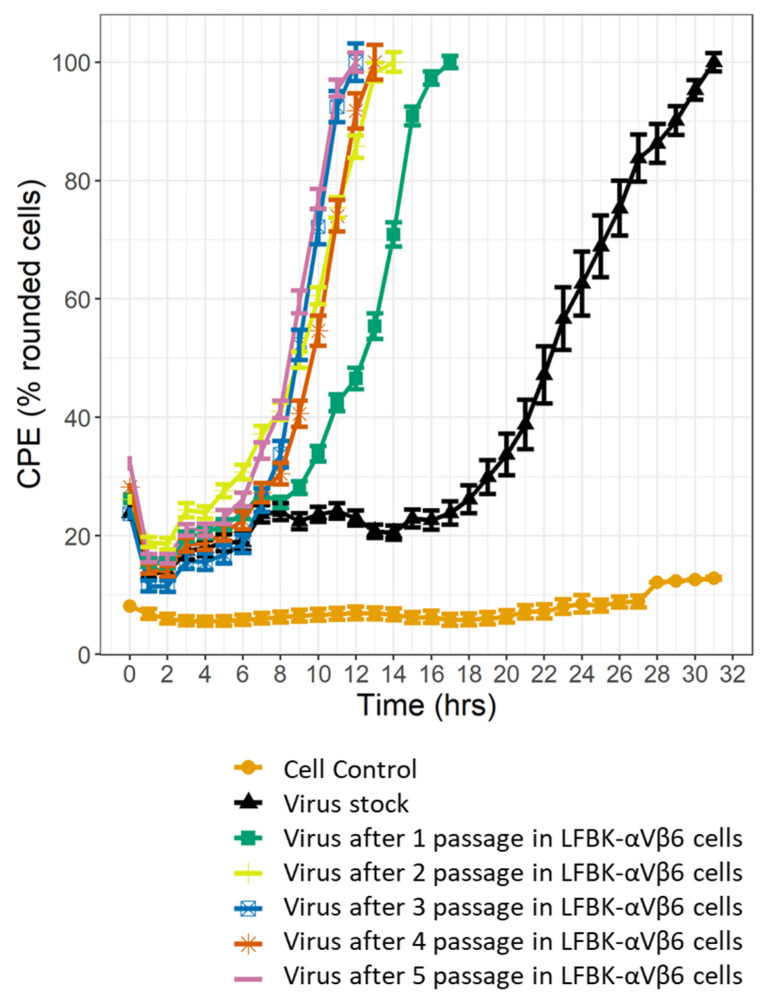
A killing curve measuring CPE (i.e., the percentage of rounded cells) caused by the IRN/22/2015 isolate passaged in LFBK-αVβ6 cells. The error bars represent the standard error or the mean (SEM) at each time point, calculated based on six replicates. The virus stock is shown in black, while passages 1–5 are show in green, yellow, blue, red and purple, respectively.

**Figure 2 viruses-14-01820-f002:**
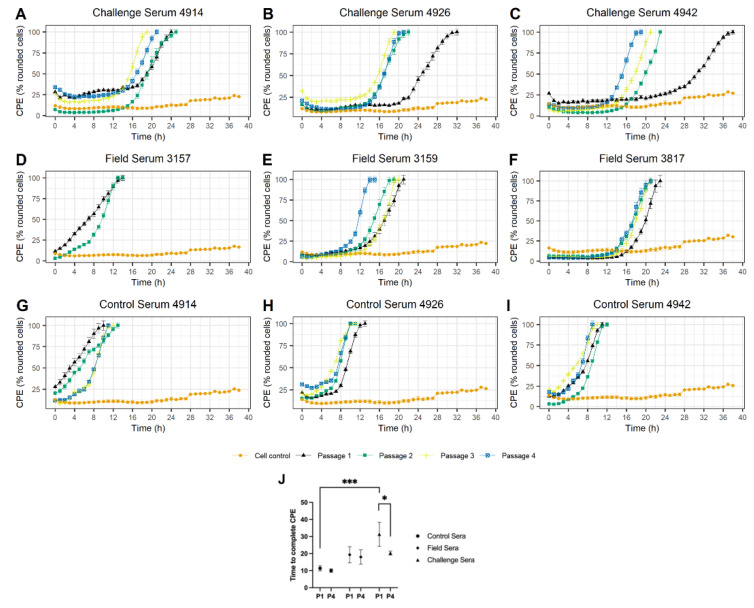
Killing curves measuring the percentage CPE (i.e., rounded cells) induced by the IRN/22/2015 isolate grown in the presence of sub-neutralising levels of sera from the Challenge group (**A**–**C**), Field group (**D**–**F**) or Control group (**G**–**I**). Each curve (black, green, yellow or blue) represents a single passage (P1, P2, P3 or P4, respectively). LFBK-αVβ6 only controls are shown in orange. Error bars represent the SEM calculated based on six replicates. Statistical comparison of time to CPE upon the initial and the final passages of the virus in presence of sera from Control, Field, and Challenge group (**J**). Markers represent mean and bars represent SEM. Statistical significance was indicated: * *p* < 0.05, *** *p* < 0.001.

**Figure 3 viruses-14-01820-f003:**
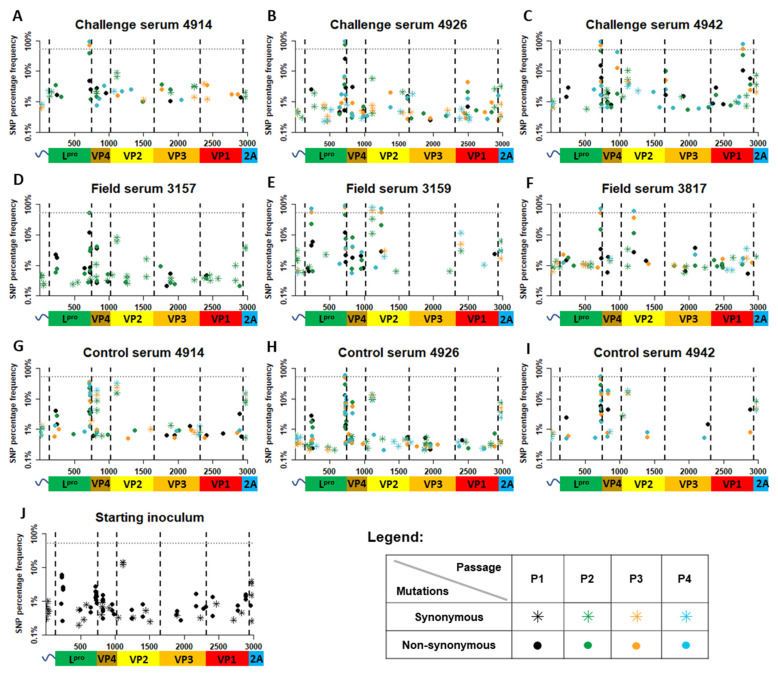
Mutations identified in virus populations passaged in the presence of FMDV sub-neutralising sera and in control viruses. Viral populations which were passaged in presence of Field sera (**A**–**C**), Challenge sera (**D**–**F**) or Control sera (**G**–**I**), as was described for Figure 2, were sequenced using Illumina MiSeq. All mutations which were detected above error threshold and in both sequencing duplicates (as specified in Material and Methods) (except Field Serum 3817, for which one technical replicate failed to sequence) were mapped along the amplicon, which spanned from the terminal section of the 5′ UTR to the region encoding the 2A protein. Synonymous mutations are represented by asterisks, while non-synonymous mutations are represented by filled circles. Both symbols were colour-coded according to the passage, with black, green, orange and turquoise representing P1, P2, P3 and P4, respectively. Starting inoculum (**J**), i.e., IRN/22/2015 isolate adapted to LFBK-αVβ6 cells over five passages, was sequenced with each individual MiSeq sequencing run (*n =* 3). Mutations detected on all three MiSeq runs were included, with synonymous mutations marked with an asterisk and non-synonymous mutations marked with a filled circle. The grey dashed horizontal line indicates the 50% threshold used to determine consensus changes.

**Table 1 viruses-14-01820-t001:** Accumulation of nucleotide substitutions which were either specific to treatment with FMDV sub-neutralising sera or increased upon passage in the presence of sub-neutralising sera, which, once detected, persisted in all subsequent passages *.

				Passage			
Protein	Amino Acid Substitution	Serum Sample	^+^ Starting Inoculum	1	2	3	4
Lpro	K29R	Control 4914	5.63%	4.10%	ND	ND	ND
		Control 4926		2.75%	1.72%	0.42%	0.34%
		Control 4942		2.47%	ND	ND	0.50%
		Field 3157		2.23%	0.60%		
		Field 3159		4.45%	22.94%	55.98%	74.39%
		Field 3817		1.46%	ND	ND	ND
		Challenge 4914		3.52%	3.43%	ND	ND
		Challenge 4926		2.53%	ND	ND	ND
		Challenge 4942		1.42%	ND	ND	ND
	N189D	Challenge 4926	ND	0.53%	3.64%	5.92%	6.73%
	W192R	Control 4914	1.96%	3.81%	22.40%	35.12%	31.14%
		Control 4926		12.74%	29.63%	49.74%	58.65%
		Control 4942		5.84%	28.46%	44.14%	53.30%
		Field 3157		12.18%	52.28%		
		Field 3159		12.06%	46.39%	80.50%	93.84%
		Field 3817		3.50%	15.31%	52.20%	71.21%
		Challenge 4914		4.87%	38.63%	71.33%	93.92%
		Challenge 4926		25.36%	73.50%	91.96%	97.07%
		Challenge 4942		15.57%	47.17%	69.96%	91.74%
VP4	L71P	Field 3159	ND	ND	1.50%	ND	2.74%
		Challenge 4926		ND	ND	0.49%	0.28%
		Challenge 4942		ND	1.83%	12.64%	42.07%
VP2	S37	Control 4914	13.17%	15.58%	15.76%	23.68%	32.35%
		Control 4926		9.23%	13.61%	11.30%	9.59%
		Control 4942		15.39%	18.62%	15.65%	18.41%
		Field 3157		8.08%	6.13%		
		Field 3159		10.93%	33.43%	62.52%	78.53%
		Field 3817		3.37%	0.93%	ND	ND
		Challenge 4914		8.90%	6.53%	ND	ND
		Challenge 4926		5.68%	0.41%	0.75%	ND
		Challenge 4942		10.38%	5.28%	4.58%	3.11%
	L66F	Field 3817	ND	2.76%	11.35%	35.70%	60.95%
	K80E	Field 3157	0.44%	ND	0.28%		
		Field 3159		2.84%	21.08%	54.08%	71.96%
VP3	P4S	Challenge 4942	ND	1.69%	9.89%	5.07%	2.48%
VP1	Q58H	Challenge 4926	ND	0.68%	2.11%	4.33%	1.28%
	A155T	Field 3817	ND	ND	ND	ND	1.68%
		Challenge 4942		10.49%	34.44%	52.37%	76.38%

***** A nucleotide substitution was required to match one of the following criteria: (1) be absent in the starting inoculum and the Control sera but detected in a viral population grown in the presence of FMDV sub-neutralising sera and in the subsequent passages (i.e., if a substitution appeared in passage x, it was required to remain at passage x + 1, x + 2 etc.). Substitutions which “skipped” any subsequent passages were considered as not fixed in the viral population; (2) be detected in the starting inoculum and/or Control sera at low frequencies and drastically increase in frequency over passages in presence of FMDV sub-neutralising sera. Again, once detected, such substitution was required to remain in any subsequent passages. To prevent biased reporting, once a substitution was detected in a viral population grown in the presence of FMDV sub-neutralising sera, its presence in other samples was also reported. In case of substitutions where other samples are also reported, the fixed substitutions had their frequency values underscored **^+^** An average percentage of the three sequenced starting inoculum samples. ND—not detected.

## Data Availability

Raw Illumina reads generated for this study have been submitted to GenBank under the BioProject accession number SUB11548315.

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
