# Peer review of "Establishing an In Vitro System to Assess How Specific Antibodies Drive the Evolution of Foot-and-Mouth Disease Virus"

_viruses, 2022, doi:10.3390/v14081820_

Round 1
Reviewer 2 Report
The authors have responded well to previous comment about the need for demonstrating that the mutations are not the result of the virus adapting to the LF-BK αVβ6 cell culture. The article is interesting and a unique take on evaluating how a virus swarm may adapt to sub-neturalizing antibody conditions. I have the following comments/suggestions:
- The differing X-axis lengths of Figure 2 A through I mask the efficacy of the results as it gives the impression of similar length of times to CPE upon looking at the figure while there are rather dramatic differences. It would be better to present the data in Figure 2A through 2I in the same way as the average data is presented in Figure 2J. This would allow for a standard axis and make it easier for the reader to grasp the data set upon looking at the figure.
- Figure 4 is orientated incorrectly with it visually flowing from Figure 4B to Figure 4A then to Figure 4C. The authors present a square on Figure 4A roughly covering the area of a pentamer that is then examined slightly zoomed in as parts of Figure 4B and 4C. This would be far better served by having a black pentamer as opposed to a black square. While it is an attractive figure it adds very little to the manuscript due the scale it is presented. It is referenced twice in the text and in both instances the discussion is about the changes in amino acids and/or side chain lengths which can not be observed in Figure 4. It would probably be best to either delete Figure 4 or significantly rework it so that it displays the discussed amino acids on a level in which the side chains can be observed in the figure, such as a trimer or a close up of a junction site between trimers.
- Line 423 refers to VP230 being both a consensus and near-consensus mutation but it is not mentioned anywhere else, specifically Table 1 like the other mutations mentioned in the sentence.
- I found the inclusion of the Lpro mutation information must interesting especially residue W192. It is worth noting in the article that residue W192 within Lpro is a part of the substrate binding pocket and discussed in previous structural publications (Guarne et al 1998 doi: 10.1093/emboj/17.24.7469). This includes the discussion that a Tryptophan at residue 192 enhances self-processing compared to other viral Lpro’s. As such the mutation presented here might reduce such self-processing or influence substrate binding efficiency.
- In regards to the high abundance of W192R what the authors may be observing is continued adaptation to the cell line but in a means that is not reflected by enhanced CPE. I do not believe that this alters any conclusions about the emergence of mutations in the structural proteins but may be worth noting in the article in the paragraph discussing the Lpro mutations in the discussion (lines 587 to 600).
- The sentence from 587 to 589 is confusedly written. The terms non-synonymous but found in all samples seems contradictory. Furthermore N189D is listed on Table 1 as only occurring in Challenge 4926, far from all samples while W192R was found in all samples and appears to be the most dramatically increasing mutation of the dataset.
Minor
- Figure 1: While the figure caption refers to the cells as LFBK-αVβ6 the key within the figure calls the LFBK cells. Suggest correcting LFBK is the designation of original cell line not expressing the integrin
- Line 283: Remove an extra period
- Line 289: Place space between period and Overall
- Line 340-342 appears to be an incomplete sequence. It starts with Additionally but doesn’t say what this is in addition to
- Line 389: word duplication ”were were”
- Line 422: It might be a good to define what low frequency is at the end of the sentence
- Table 1: Some numbers have % others do not, this should be standardized
- Line 441: There appears to be an extra space between isolate and was
- Lines 450 through 455: some amino acid names are capitalized others are not, please standardize
- Line 509: delete “which” before “sufficient only” at end of sentence
Author Response
Please see the attachment

This manuscript is a resubmission of an earlier submission. The following is a list of the peer review reports and author responses from that submission.
Round 1
Reviewer 2 Report
The article is well written and a very interesting approach with high potential for future usage in evaluation of the role FMD vaccines may play in the emergence of new strains and escape variants. There are a number of very interesting possibilities for the application of this procedure.
The conclusions are well presented and interpreted in the context of structural changes to the FMDV capsid and how that would relate to antibody binding.
Major Concerns:
If a mutation was introduced that allowed for better replication or infection of the LF-BK aV/b6 cell line I believe it would present the same result of enhanced and quicker CPE while also becoming more dominant within the viral population. The authors do not present experimental data that the virus swarm did not adapt for better replication in the the LF-BK aV/b6 cell line. I would like to see a control experiment that accounts for this possibility.
While Figure 1 does successfully demonstrate that the time to complete CPE is decreasing with successive passages outside of 4926 and 4942 the differences are hard to see due to the scales used. Figure 1 also demonstrates that the time to full CPE is decreasing on the control samples. While not to the same extent as in 4936 and 4942 the difference does not look as all that different from 4914, 3817, or 3157, which should be under antibody selective pressure, this may be a result of the scales used in Figure 1. This goes back to my concerns that the article needs a control to demonstrate that the final P4 viral swarm is avoiding an antibody response and not adapting to better replicate in the cell line.
The authors entirely focus on changes in the structural proteins and no mention is made if there were any changes in the non-structural regions. If all the changes were localized in the structural proteins please state this, if true it would strengthen the argument that this is an escape of neutralizing antibodies and not an adaptation to replication in the cell line. If mutations are present in the non-structural regions in depth evaluation is not needed but a statement about whether or not mutations were present and a list of percent changes for each region would be greatly beneficial.
Where the control passed viruses sequenced and did they demonstrate any of the same mutations as those under sub-neutralizing pressure?
Minor Edits:
The abbreviation FMD is used in the abstract (line 15) but is not defined in the abstract.